# EFFICIENT QUAD BAYER DEMOSAICING WITH LOOK-UP TABLES

## ABSTRACT

This paper presents a novel look-up table-based demosaicing method for quad Bayer pattern. Recent deep learning approaches, though effective, are computationally expensive and unsuitable for efficient implementation. We first introduces a residual LUT to reduce memory and computational complexity. The method employs a two-stage demosaicing process: the first stage performs primary demosaicing, while the second stage enhances high-frequency components. Each stage consists of a number of residual LUTs and they are stacked both in serial and parallel to effectively enlarge the receptive field size. In experiments, our method achieves stable performance and improved image quality with efficiency. In addition, the size and the computational complexity of our method enable efficient hardware implementation.

## 1 INTRODUCTION

In a digital camera such as a smartphone, the raw image data captured by the camera image sensor (CIS) is converted into an RGB image, which is more familiar to human perception, through the use of an image signal processor (ISP) that consists of various processing modules. In CIS, a color filter array (CFA), which is spatially sampled for each pixel, generates the color-mosaiced raw data by allowing only light of specific wavelength ranges to pass through each pixel. The mosaiced raw data is processed using an ISP to convert it into RGB format, making it perceivable by humans. The primary processing module in this step is demosaicing. The demosaicing reconstructs the missing color information that occurs during light-source sampling by a CFA. This process converts a single-channel raw image into an RGB 3-channel image. Demosaicing is crucial for the performance

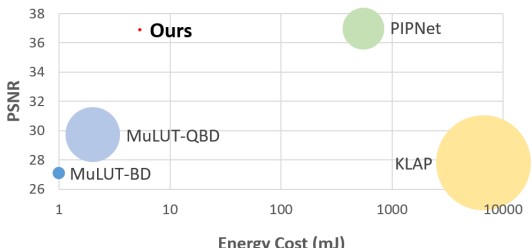

Figure 1. Energy cost and PSNR comparison on McMaster benchmark for the quad Bayer demosaicing task. The area of the circle represents the size of the model. Our method achieves a good performance even at a very small LUT size (73KB), which is small enough to be integrated into smartphone hardware.

of the ISP because it determines the fundamental image quality, such as sharpness, and can introduce various artifacts, including color artifacts, Moiré patterns, false direction, zipper artifacts, and jagging.

As the demand of smartphone users for high-quality images has increased, the pixel size of CIS has been reduced to allow CFA to sample more spatially finely, enabling the achievement of high-resolution images. However, due to the reduction in pixel size, the drawback of poor signal-to-noise ratio (SNR) has been exacerbated in low-light environments. To address this issue, most smartphone manufacturers have adopted the use of a 4x4 quad Bayer pattern, which allows for more convenient binning to increase SNR compared to the traditional 2x2 Bayer pattern.

Although the quad Bayer pattern has these advantages of improving the SNR for high-resolution sensors, it results in increased difficulty in demosaicing. Theoretically, the 4x4 quad Bayer pattern is spatially sampled twice as finely compared to the 2x2 Bayer pattern. This means that the number of filter taps required for demosaicing in the quad Bayer pattern is twice as many as that in the Bayer

pattern. Since most smartphone manufacturers utilize hardware-based ISPs, the number of filter taps has a crucial impact on hardware complexity.

Recently, various studies on quad Bayer demosaicing using a deep-learning approach have emerged (Kim et al., 2019; A Sharif et al., 2021; Zheng et al., 2024; Zeng et al., 2024; Xu et al., 2024; Sharif et al., 2021; Lee et al., 2023). However, most of the research utilizes numerous convolution layers or transformer architectures. These approaches demonstrate remarkable performance. However, they require a significant amount of computation using high-end GPUs. Consequently, these approaches are not suitable for implementation on smartphones or edge devices. Recently, ISPs for smartphones should provide the capability for high-resolution video recording (*e.g.* 8K@30FPS video). Thus, an implementable approach on hardware is necessary for commercialization.

To reduce computational complexity, several look-up table (LUT)-based approach has been introduced (Jo & Kim, 2021; Ma et al., 2022; Li et al., 2024b; Liu et al., 2023; Li et al., 2022; 2024c;a). This approach trains a deep neural network (DNN) and transfers it into the LUT to significantly reduce computation. However, the memory utilization of the LUT is exponentially proportional to its input dimension. As a result, it was difficult to apply to the demosaicing task that requires a large receptive field. In this paper, we introduce a LUT-based demosaicing method for a quad Bayer pattern CFA sensor for the first time. Although this approach improves image quality compared with traditional image processing techniques, it significantly reduces the computational load affordable for hardware ISP implementation. For the quad Bayer demosaicing task, we effectively enlarge the receptive field by serially and parallelly cascading multiple LUTs. Additionally, we incorporate a directed chroma interpolation method, widely used in the traditional demosaicing techniques, into the overall architecture to ensure stable demosaicing performance.

## 2 RELATED WORK

### 2.1 QUAD BAYER DEMOSAICING

A number of deep learning-based methods have been proposed to enhance the performance of the traditional Bayer pattern demosaicing (Gharbi et al., 2016; Tan et al., 2017; Kokkinos & Lefkimmiatis, 2018; Tan et al., 2018; Qian et al., 2019; Liu et al., 2020; Xing & Egiazarian, 2022). Based on these successes and the emergent of quad Bayer pattern image sensors, researchers have also applied deep learning approaches to the quad Bayer pattern demosaicing (Kim et al., 2019; 2021; A Sharif et al., 2021; Zheng et al., 2024; Zeng et al., 2024; Jia et al., 2022; Wu et al., 2022; Xu et al., 2024).

DPN (Kim et al., 2019) applied a U-Net (Ronneberger et al., 2015) like feature pyramid network structure with residual connections. Based on the U-Net structure, PIPNet (A Sharif et al., 2021) leveraged depth attention (Hu et al., 2018) and convolutional spatial attention (Woo et al., 2018). In addition, two perceptual losses, a VGG19 feature loss and a CIEDE2000-based color loss (Luo et al., 2001), were applied to further enhance the visual quality of the quad Bayer demosaicing task. DRNet (Zheng et al., 2024) adopted multi-scale encoder-decoder architecture with a kind of self-attention (Vaswani et al., 2017) mechanism. DJRD (Zeng et al., 2024) integrated Swin-Transformer (Liu et al., 2021) and multi-scale wavelet transform to capture non-local dependencies, frequency and location information effectively. It employed the HDR-VDP2 visual metric (Mantiuk et al., 2011; Gharbi et al., 2016) to specifically identify and reduce Moiré and zipper artifacts. DemosaicFormer (Xu et al., 2024) introduced a multi-scale gating module to allow the interaction of cross-scale feature information.

In recent years, several studies have emerged on demosaicing algorithms for nona-Bayer pattern (Sharif et al., 2021) and Q×Q Bayer pattern (Cho et al., 2023), as the commercialization of these image sensors becomes feasible. Moreover, there has been a study on a unified demosaicing algorithm capable of simultaneously handling all those Bayer patterns within a single model (Lee et al., 2023).

The aforementioned deep learning-based demosaicing methods have achieved performance improvements by focusing on improving the deep learning models themselves. This approach is reasonable since the models generally possess sufficient model capacity and receptive fields. However, the method proposed in this paper is a highly lightweight and constrained one using LUTs, resulting in very limited model capacity and receptive field. In this situation, it is not allowed to increase the number of layers or utilize attention modules. Instead, each component of the proposed method

must be designed to maximize its effectiveness within the model's capacity constraints. To achieve this, we carefully design the model components by inheriting techniques commonly used in classical demosaicing methods (Hirakawa & Parks, 2005; Zhang & Wu, 2005; Jeon & Dubois, 2012; Kiku et al., 2016) before deep learning era. Specifically, a directed processing and a chroma processing are applied, ensuring that appropriate processing is effectively carried out according to the structure of the input image.

## 2.2 DEEP LEARNED LUT

SR-LUT (Jo & Kim, 2021) was the first to demonstrate the concept of converting a learned deep network with a restricted receptive field into a LUT for the super-resolution task. Subsequent to this work, various methods including MuLUT (Li et al., 2022; 2024b), SP-LUT (Ma et al., 2022), and RCLUT (Liu et al., 2023) have emerged by leveraging learned LUTs for a better super-resolution quality. The rationale behind employing deep networks with restricted receptive fields lies in the fact that the size of LUT being converted increases exponentially with the size of the receptive field. Consequently, aforementioned LUT-based methods limited the receptive field size to a maximum of 4 pixels. To address this, various techniques, such as a rotational ensemble (Jo & Kim, 2021), a cascading (Li et al., 2024b; Ma et al., 2022), and a decoupling (Liu et al., 2023), were introduced to effectively increase the receptive field size more than 4 pixels.

These methods focused on isotropically expanding the receptive field size for the super-resolution task. In contrast, for the demosaicing task, an effective direction-aware processing in accordance with the input image structure is important (Hirakawa & Parks, 2005; Zhang & Wu, 2005; Jeon & Dubois, 2012; Kiku et al., 2016), thereby requiring the receptive field to be expanded in a direction-aware manner. In this paper, a novel kernel design tailored to specific directions is proposed, as well as a two-stage processing approach to consider a larger number of pixels.

Notably, conventional methods typically employ LUTs size exceeding 1MB. This is typically small size, however, it can be challenging to implement on edge devices or in hardware. To resolve this, techniques aimed at reducing the size of LUTs have been developed. DFC (Li et al., 2024c) proposed a method that prioritizes the compression of the diagonal elements of the LUT and TinyLUT (Li et al., 2024a) suggested to decompose the correlated pixels, achieving a high compression rate. On the other hand, this paper proposes the concept of residual LUT, which effectively enables the reduction of the receptive field size by -1. This enables the construction of the entire pipeline using LUTs with 3-pixel and 2-pixel receptive fields, rather than using the widely used 4-pixel receptive field LUTs, thus achieving a significant reduction in model size.

## 3 METHOD

### 3.1 RESIDUAL LUT

Firstly, we propose a novel residual LUT concept, which processes the residual value and subsequently compensates for it, inspired by residual learning (He et al., 2016). Previously, a deep network with a 4-pixel receptive field could be converted into a 4D LUT. In contrast, the proposed residual LUT transforms the 4-pixel input into a residual form as

$$V = LUT[p_1 - p_0][p_2 - p_0][p_3 - p_0] + p_0. \tag{1}$$

The application of the residual LUT enables the reduction of a 4D LUT to a 3D LUT, and a 3D LUT to a 2D LUT, resulting in a substantial decrease in the overall LUT size. For instance, SR-LUT-S (Jo & Kim, 2021), which originally had a size of 1.3MB, can be reduced to 77KB. Furthermore, the conversion of a 4D LUT to a 3D LUT also leads to a reduction in computational complexity because 4D interpolation required for output generation is also changed to 3D interpolation. The proposed residual LUT plays a essential role in reducing our LUT size and computational complexity, particularly in the case of chroma blocks, as will be discussed later.

### 3.2 OVERVIEW

The overview of our method is shown in Fig. 2. The proposed method consists of two stages. In the first stage, a primary demosaicing is applied to the quad Bayer input image to generate a primary

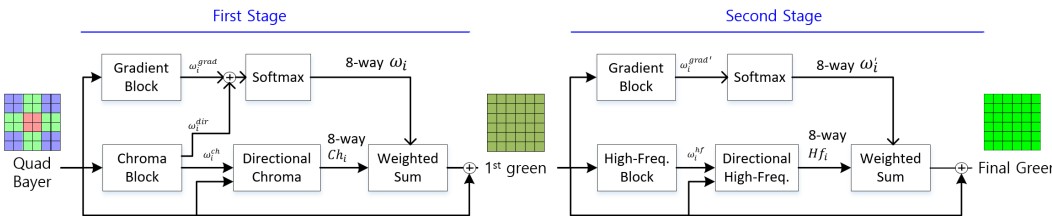

Figure 2. Our methodology comprises two distinct stages. The first stage performs initial demosaicing, followed by high-frequency enhancement in the second stage. The type of all LUTs in our method is the proposed residual LUT, ensuring memory efficiency.

demosaiced image. The second stage takes the result of the primary stage as its input and performs a secondary demosaicing. Since the first stage involves restoring color values from other color pixels and distant pixels, it is challenging to achieve a good level of restoration in a single step, particularly when the model capacity is small, as is the case with the proposed method in this paper. To alleviate this, the secondary demosaicing is employed to enhance the results of the primary demosaicing, thereby enhancing the final quad Bayer demosaicing performance.

Each stage comprises a chroma block and a gradient block. The chroma block extracts chroma components in eight directions: up, down, left, right, up-left, up-right, down-left, and down-right. In the gradient block, the gradient values of the input quad Bayer image are calculated in the eight directions, and the weights for each direction are then computed based on the gradient values. Finally, the output demosaiced image is generated by taking the weighted sum of the chroma components and the weights, and the input quad Bayer image is added through a residual connection.

Conventional LUT-based approaches primarily target super-resolution (SR) and denoising tasks, but they are difficult to apply to demosaicing tasks, which require a wider receptive field. The proposed method is designed to widen the receptive field by connecting small-sized LUTs in series and in parallel, enabling the reference of distant pixel information. This characteristic allows the method to be successfully implemented for the quad Bayer demosaicing task for the first time. Furthermore, by integrating directed processing and chroma processing techniques, which are commonly used in the demosaicing field (Hirakawa & Parks, 2005; Zhang & Wu, 2005; Jeon & Dubois, 2012; Kiku et al., 2016), into a learning-based LUT, the performance is improved within the limited model capacity.

The following explanation demonstrates how to generate a green pixel at the top-left corner location of the red pixel group. Green pixel values in the other corners (*i.e.* top-right, bottom-left, and bottom-right) can be generated within the same manner. We utilize a conventional, non deep learning-based quad Bayer interpolation technique for the generation of red and blue pixels for the final color image.

We present our first demonstration on the green channel only in this study. This is because the green channel (luminance) is considered as a key in estimating other missing color samples due to several reasons – 1) the green channel has twice as many samples as red or blue channels in the (quad) Bayer mosaic pattern, and 2) the human visual system is most sensitive to the green wavelength (Zhang & Wu, 2005). However, it is worth noting that, our method can also be applied for the generation of red and blue pixels with appropriate adjustments.

### 3.3 First Stage

**Chroma Block**   A chroma block produces chroma components in eight directions from quad Bayer input image. For a direction-aware processing, three sets of LUT operations are employed consisting of horizontal, vertical, and diagonal configurations (Fig. 3). The horizontal configuration generates outputs for the left and right directions, while the vertical configuration generates outputs for the up and down directions. The diagonal configuration generates outputs for the remaining four diagonal directions: up-left, up-right, down-left, and down-right. Each of these configuration consists of four LUTs, which accept a predefined pixel region as input and produce weights $w^{ch}$ for synthesizing directional chroma components, as well as direction weights $w^{dir}$ for combining the resulting chroma components across eight directions.

For instance, as illustrated in Fig. 3a, four LUTs in the horizontal configuration are designed to take horizontal directionality into account. The LUT *ChromH_L* which is responsible for processing

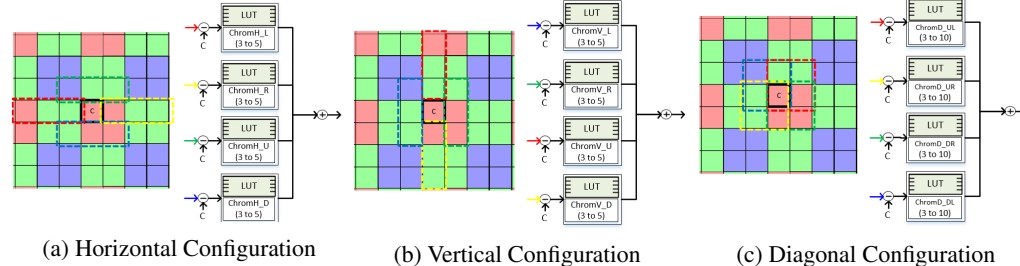

(a) Horizontal Configuration    (b) Vertical Configuration    (c) Diagonal Configuration

Figure 3. Three different directional configurations in the chroma block of the first stage, allowing better consideration of the image structures.

the left-directed pixels receives three pixels colored by the red box as input in the form of the residuals with respect to the central pixel $C$ ($LLL - C$, $LL - C$, and $L - C$), and generates three weights ($w_{l_l}^{ch}$, $w_{r_l}^{ch}$, and $w_{\hat{r}_l}^{ch}$) for synthesizing left and right chroma outputs, as well as two direction weights ($w_{l_l}^{dir}$ and $w_{r_l}^{dir}$) for combining the left and right chroma components. In other words, $w_{l_l}^{ch}, w_{r_l}^{ch}, w_{\hat{r}_l}^{ch}, w_{l_l}^{dir}, w_{r_l}^{dir} = ChromH\_L[LLL - C][LL - C][L - C]$.

Similarly, three right-directed pixels colored by the yellow box are input to the LUT *ChromH_R* in the form of the residuals and generates total five weights as $w_{l_r}^{ch}$, $w_{r_r}^{ch}$, $w_{\hat{r}_r}^{ch}$, $w_{l_r}^{dir}, w_{r_r}^{dir} = ChromH\_R[RRR - C][RR - C][R - C]$. After conducting the same process for the remaining LUTs *ChromH_U* and *ChromH_D*, then we can obtain the final horizontal weight components as $w_l^{ch} = \sum_i w_{l_i}^{ch}$, $w_r^{ch} = \sum_i w_{r_i}^{ch}$, $w_{\hat{r}}^{ch} = \sum_i w_{\hat{r}_i}^{ch}$, $w_l^{dir} = \sum_i w_{l_i}^{dir}$, and $w_r^{dir} = \sum_i w_{r_i}^{dir}$ for $i \in \{l, r, u, d\}$.

The same processing occurs in both the vertical configuration (Fig. 3b) and the diagonal configuration (Fig. 3c), with the input pixel locations varying accordingly. In the vertical case, total five weights are generated ($w_u^{ch}$, $w_d^{ch}$, $w_{\hat{d}}^{ch}$, $w_u^{dir}$, and $w_d^{dir}$). In the diagonal case, total ten weights are generated ($w_{ul_u}^{ch}$, $w_{ul_l}^{ch}$, $w_{ur}^{ch}$, $w_{dl}^{ch}$, $w_{dr_d}^{ch}$, $w_{dr_dr}^{ch}$, $w_{dr_r}^{ch}$, $w_{\hat{dr}_d}^{ch}$, $w_{\hat{dr}_r}^{ch}$, and $w_{\hat{dr}_{dr}}^{ch}$) but without direction weights due to a limited receptive field diagonally.

The computed weights $w^{ch}$ are then utilized to synthesize two horizontal ($Ch_l$ and $Ch_r$), two vertical ($Ch_u$ and $Ch_d$), and four diagonal ($Ch_{ul}$, $Ch_{ur}$, $Ch_{dl}$, and $Ch_{dr}$) directional chroma components, in a total of eight, as follows:

$$Ch_l = w_l^{ch}(LL - C) + (1 - w_l^{ch})(L - C), \quad Ch_r = w_r^{ch}(RR - C) + (1 - w_r^{ch})(\hat{R}_r - C),$$

$$\text{where } \hat{R}_r = w_{\hat{r}}^{ch}(RR - R) + R, \tag{2}$$

$$Ch_u = w_u^{ch}(UU - C) + (1 - w_u^{ch})(U - C), \quad Ch_d = w_d^{ch}(DD - C) + (1 - w_d^{ch})(\hat{D}_d - C),$$

$$\text{where } \hat{D}_d = w_{\hat{d}}^{ch}(DD - D) + D, \tag{3}$$

$$Ch_{ul} = w_{ul_u}^{ch}(U - C) + w_{ul_l}^{ch}(L - C), \quad Ch_{ur} = w_{ur}^{ch}(UR - C), \quad Ch_{dl} = w_{dl}^{ch}(DL - C),$$

$$Ch_{dr} = w_{dr_d}^{ch}(\hat{D}_{dr} - C) + w_{dr_r}^{ch}(\hat{R}_{dr} - C) + w_{dr_{dr}}^{ch}(\hat{DR}_{dr} - C),$$

$$\text{where } \hat{D}_{dr} = w_{\hat{dr}_d}^{ch}(DD - D) + D, \quad \hat{R}_{dr} = w_{\hat{dr}_r}^{ch}(RR - R) + R,$$

$$\hat{DR}_{dr} = w_{\hat{dr}_{dr}}^{ch}(DDR - DR) + (1 - w_{\hat{dr}_{dr}}^{ch})(DRR - DR) + DR, \tag{4}$$

and please refer to Fig. 5a for the referenced pixel locations. We exploit a blending of a far chroma (*e.g.* $LL - C$) and a near chroma (*e.g.* $L - C$), which were widely used in traditional demosaicing algorithms, designed to facilitate the effective processing of color values (Hirakawa & Parks, 2005; Zhang & Wu, 2005; Jeon & Dubois, 2012; Kiku et al., 2016). Here we synthesize several pseudo pixels such as $\hat{R}$s, $\hat{D}$s, and $\hat{DR}$, given that the quad Bayer pattern does not contain any green pixels in these locations. This can be regarded as a synthesis of near chroma values from far chroma values, and we empirically found an improved outcome can be obtained using the pseudo pixels. The direction weights $w^{dir}$s are subsequently merged with the outputs produced by a gradient block explained in the subsequent chapter, and the resulting weight is then employed.

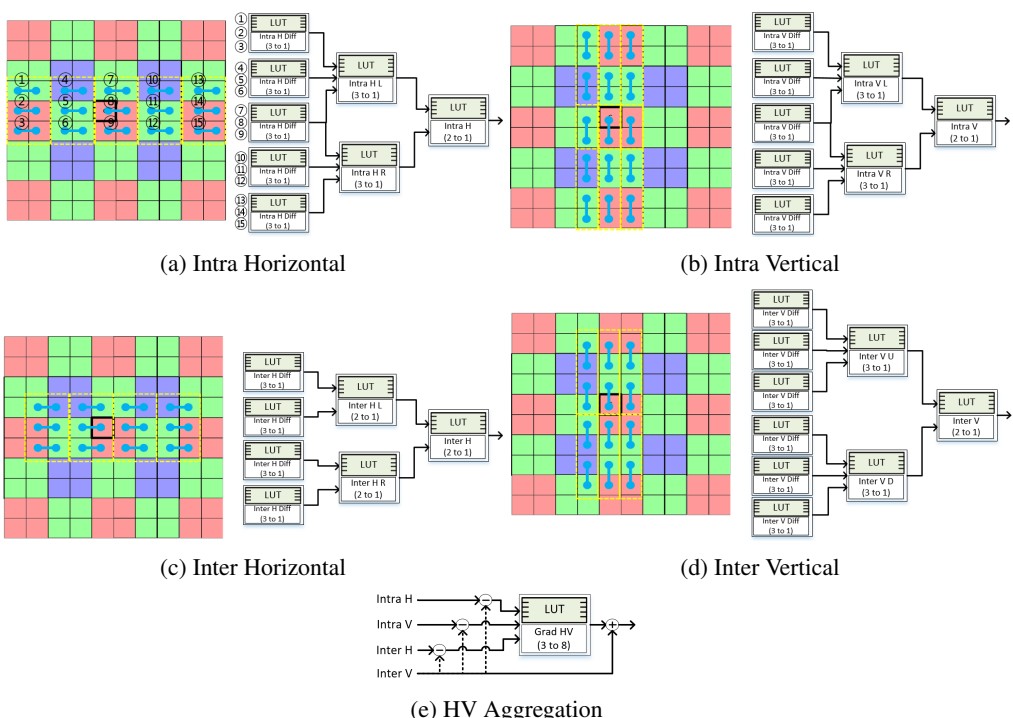

(a) Intra Horizontal (b) Intra Vertical

(c) Inter Horizontal (d) Inter Vertical

(e) HV Aggregation

Figure 4. The horizontal and vertical processing in the gradient block of the first stage. Compute the gradient values between intra and inter channels first and use them as inputs for the LUTs.

**Gradient Block**   A gradient block is a responsible for generating the directional weights $w_i$ which is utilized in weighted averaging of the directional chroma. To provide appropriate weights based on pixel values for each directions, the gradient block detects edges by calculating gradient values between adjacent pixels. Thus, the LUTs of the gradient block use the gradient values as input instead of the intrinsic pixel values.

Utilizing more gradient values within large receptive field is more confident and suitable to treat complex edges compared to using only that of center pixels. However, the number of inputs of LUT enormous affects to the memory requirement of LUT as mentioned before. We propose a cascade structure of LUTs which effectively enables enlargement of the receptive field and improves the representation power of LUTs. By constructing LUTs in cascade, the memory requirement is extremely reduced from $\mathcal{O}(pt^N)$ to the $M * \mathcal{O}(pt^k)$. Assuming that we construct a single LUT with 15-input then the required memory is about 4.3TB (when $pt$=7). On the other hand, the structure of the cascade LUT as shown in Fig. 4a, seven 3-input LUT and a single 2-input LUT are required, which means that only 2.4KB of memory is sufficient to construct the

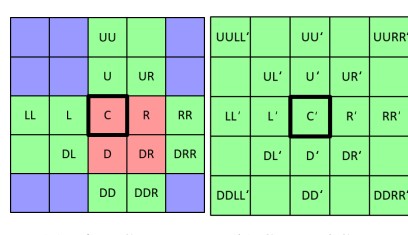

(a) First Stage (b) Second Stage

Figure 5. Pixel positions used to compute directional chroma components. Please see the text for details.

cascade structure (*i.e.* $7^{15}$ to $7 \times 7^3 + 7^2$). The cascade LUT for *Intra Horizontal* (Fig. 4a) detects the vertical edge component with horizontal gradient values. However, horizontal gradient values are acquired with only adjacent intra-color pixels, which can cause suffering from aliasing artifact due to the periodic pattern of 4-pixels. Thus, not only the gradient values on intra-color pixels but inter-color gradient should be considered together. Fig. 4c shows the component for generating *Inter Horizontal* gradients with the gradient values between inter-color pixels.

In addition to the horizontal components, vertical components *Intra Vertical* (Fig. 4b) and *Inter Vertical* (Fig. 4d) can be acquired by replacing input gradient values by vertical gradient values with the similar manner.

As shown in Fig. 4e, the horizontal and vertical components previously calculated are then aggregated to generate *HV* directional components $w_{l_{hv}}^{grad}$, $w_{r_{hv}}^{grad}$, $w_{u_{hv}}^{grad}$, $w_{d_{hv}}^{grad}$, $w_{ul_{hv}}^{grad}$, $w_{ur_{hv}}^{grad}$, $w_{dl_{hv}}^{grad}$, and $w_{dr_{hv}}^{grad}$. Here, the residual LUT structure is also used to reduce the size. Similarly, *SB* directional components $w_{l_{sb}}^{grad}$, $w_{r_{sb}}^{grad}$, $w_{u_{sb}}^{grad}$, $w_{d_{sb}}^{grad}$, $w_{ul_{sb}}^{grad}$, $w_{ur_{sb}}^{grad}$, $w_{dl_{sb}}^{grad}$, and $w_{dr_{sb}}^{grad}$ also computed using the slash and backslash gradients (Fig. 7), because *HV* alone does not consider the diagonal gradients well. Finally, the directional weights of the gradient block $w_l^{grad}$, $w_r^{grad}$, $w_u^{grad}$, $w_d^{grad}$, $w_{ul}^{grad}$, $w_{ur}^{grad}$, $w_{dl}^{grad}$, and $w_{dr}^{grad}$ are generated by simply adding the *HV* and *SB* components.

To obtain the final directional weights for for chroma averaging, two direction weights from the chroma block ($w^{dir}$s) and the gradient block ($w^{grad}$s) are merged by addition and a softmax is applied as follows:

$$w_i = \frac{e^{(w_i^{dir}+w_i^{grad})}}{\sum_j e^{(w_j^{dir}+w_j^{grad})}} \text{ for } i \in \{l, r, u, d, ul, ur, dl, dr\}. \tag{5}$$

Final output of the first stage is computed as follows:

$$C' = \frac{\sum w_i Ch_i}{\sum w_i} + C, \tag{6}$$

then the initial green output image is obtained.

### 3.4 SECOND STAGE

The second stage consists of a high-frequency block and a gradient block. The high-frequency block executes a process that is largely analogous to that of the first stage's chroma block. Since the input image has color values filled in after the processing of the first stage, the processing of the second stage can be regarded as an enhancement of directional high-frequency components. This block generates weights $w^{hf}$ and the weights are utilized to synthesize two horizontal ($Hf_l$ and $Hf_r$), two vertical ($Hf_u$ and $Hf_d$), and four diagonal ($Hf_{ul}$, $Hf_{ur}$, $Hf_{dl}$ and $Hf_{dr}$) high-frequency components as follows:

$$Hf_l = w_{ll}^{hf}(LL'-C') + w_l^{hf}(L'-C'), \quad Hf_r = w_{rr}^{hf}(RR'-C') + w_r^{hf}(R'-C'), \tag{7}$$

$$Hf_u = w_{uu}^{hf}(UU'-C') + w_u^{hf}(U'-C'), \quad Hf_d = w_{dd}^{hf}(DD'-C') + w_d^{hf}(D'-C'), \tag{8}$$

$$Hf_{ul} = w_{ul}^{hf}(UL'-C') + w_{uull}^{hf}(UULL'-C'), \quad Hf_{ur} = w_{ur}^{hf}(UR'-C') + w_{uurr}^{hf}(UURR'-C'),$$
$$Hf_{dl} = w_{dl}^{hf}(DL'-C') + w_{ddll}^{hf}(DDLL'-C'), \quad Hf_{dr} = w_{dr}^{hf}(DR'-C') + w_{ddrr}^{hf}(DDRR'-C'), \tag{9}$$

and please refer to Fig. 5b for the referenced pixel locations.

In this block, the directional weights $w^{dir}$s are no longer generated, instead the gradient block solely responsible for generating them. This is a reasonable design choice as the gradient block can consider more pixels with directionality compared to this block. The gradient block of the second stage is exactly the same as that of the first stage.

### 3.5 STEPWISE FINETUNING

After training the DNN with the above structure, the values at predetermined sampling point positions are converted into LUTs. During inference using the LUT, output values are generated through interpolation based on the sampling points (Jo & Kim, 2021). This process introduces interpolation errors, which are mitigated through sampling point-aware finetuning (Li et al., 2022; 2024b).

Since the proposed method consists of multiple stacked LUTs, we perform a stepwise finetuning according to the model's characteristics, instead of finetuning all parts simultaneously. The DNNs in the first stage are first finetuned, and then the DNNs in the second stage are finetuned. Within each stage, the finetuning of each block is performed separately, with gradient blocks being finetuned first. Specifically, we first finetune the parts at the leftmost level in Fig. 4a (5 LUTs), followed by the next level (2 LUTs), then the last level (1 LUT), and finally the aggregation stage in Fig. 4e. During

Table 1. Quantitative comparisons with other methods. Our method, incorporating the residual LUT and the cascading architecture, achieves efficient implementation while maintaining a good performance relative to its compactness. For DNN-based methods, * stands for the number of parameters and † for FLOPs instead of IOPs.

| Type | Method | Size (KB) | IOPs (G) | Energy (mJ) | Kodak | | | | McMaster | | | | Urban100 | | | |
|---|---|---|---|---|---|---|---|---|---|---|---|---|---|---|---|---|
| | | | | | PSNR | SSIM | LPIPS | DISTS | PSNR | SSIM | LPIPS | DISTS | PSNR | SSIM | LPIPS | DISTS |
| LUT | MuLUT-BD | 1223 | 1.006 | 0.93 | 28.78 | 0.849 | 0.1410 | 0.1487 | 27.07 | 0.777 | 0.1732 | 0.1859 | 27.66 | 0.878 | 0.1016 | 0.1265 |
| | MuLUT-QBD | 23735 | 2.097 | 2.03 | 31.33 | 0.921 | 0.0741 | 0.0780 | 29.71 | 0.891 | 0.0959 | 0.1140 | 30.06 | 0.931 | 0.0519 | 0.0782 |
| | Ours | 73 | 4.879 | 5.37 | 37.18 | 0.972 | 0.0210 | 0.0248 | 36.87 | 0.968 | 0.0206 | 0.0313 | 35.58 | 0.976 | 0.0136 | 0.0226 |
| DNN | PIPNet | 3.459M* | 1836† | 558 | 39.19 | 0.983 | 0.0103 | 0.0144 | 36.96 | 0.960 | 0.0194 | 0.0297 | 36.86 | 0.979 | 0.0090 | 0.0176 |
| | KLAP | 17M* | 2688† | 6641 | 33.44 | 0.941 | 0.0844 | 0.0706 | 27.82 | 0.864 | 0.1350 | 0.1106 | 30.11 | 0.926 | 0.0574 | 0.0767 |

| MuLUT-BD | MuLUT-QBD | KLAP | PIPNet | Ours | GT |
|---|---|---|---|---|---|

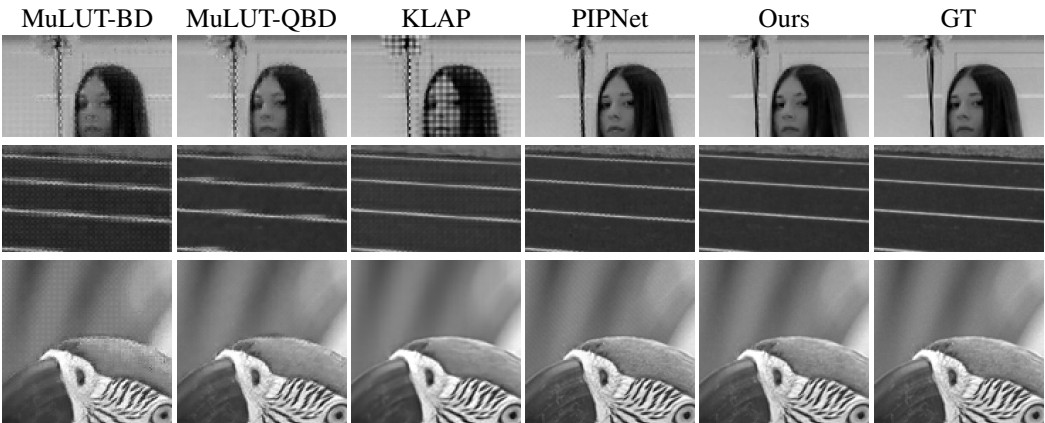

Figure 6. Please zoom in to see details for qualitative comparisons. Our method shows excellent restoration capability in thin edge areas through the directional chroma processing. It also shows less noise in flat areas.

this stepwise finetuning, all parts of the previous step are frozen. Similar procedure is applied to the remaining directions. After the finetuning of the gradient block is completed, the finetuning of the chroma block follows. The same process is repeated in the second stage, and this finetuning sequence was determined through extensive experiments.

## 4 EXPERIMENTS

### 4.1 IMPLEMENTATION DETAILS

We train our quad Bayer demosaicing deep model with the DF2K dataset (Agustsson & Timofte, 2017; Lim et al., 2017), which is widely used in the quad Bayer demosaicing task. The DF2K dataset contains 3450 RGB images with various scenery. For the training, quad Bayer pattern is sub-sampled from the RGB images with 48×48 patch size. Our model is trained with the AdamW optimizer (Loshchilov & Hutter, 2017) using the mean-squared error loss. The first stage is trained for 82K iterations with a mini-batch size of 16, and the second stage is trained for another 40K iterations. After selecting the uniform sampling points as 7, the stepwise finetuning is conducted for 20K more iterations, for each step takes 2K iterations.

After the training is completed, each part of the model is converted into its respective LUT. Once the LUTs are created, the inference can be performed using solely LUTs. The output of each LUT is obtained with tetrahedral or triangular interpolation (Jo & Kim, 2021) for 3-input LUTs and 2-input LUTs respectively. The output values are passed as input values to the next LUT, or generate the output image at the end of the process.

## 4.2 COMPARISONS WITH OTHERS

We evaluate our method with Kodak (Li et al., 2008), McMaster (Zhang et al., 2011), Urban100 (Huang et al., 2015), BSD100 (Martin et al., 2001), and MIT Moire (Gharbi et al., 2016). We report peak signal-to-noise ratio (PSNR) and structural similarity index (SSIM) (Wang et al., 2004) for the image fidelity, LPIPS (Zhang et al., 2018) and DISTS (Ding et al., 2020) for the perceptual quality, and we calculate the LUT size, the number of integer operations (IOPs), and the theoretical energy cost which is introduced in (Li et al., 2024b), for the efficiency comparisons.

We compare our method with LUT-based Bayer demosaicing method MuLUT (Li et al., 2024b), and DNN-based quad Bayer demosaicing methods PIPNet (A Sharif et al., 2021) and KLAP (Lee et al., 2023). Since MuLUT was not designed for quad Bayer demosaicing task, therefore, we test two versions: the Bayer demosaicing model proposed in the MuLUT paper (MuLUT-BD), and a customized model for the quad Bayer demosaicing task (MuLUT-QBD). MuLUT-BD consists of two-level structure: MuLUT-S followed by MuLUT-SDY and the receptive field is 5×5. For MuLUT-QBD, we replace MuLUT-S as MuLUT-SDYEHO in the MuLUT-BD so the receptive field is enlarged to 11×11.

**Quantitative Comparisons** Quantitative comparisons with other methods are shown in Table 1. The PSNR and SSIM values are calculated in the G-channel only, and the IOPs and energy costs are calculated for 1280×720 input image size.

Compared to the MuLUT variants, our method shows a significant performance gain in terms of image quality, achieving average improvement of 4.68dB across the five benchmarks. In addition the LUT size of our method is drastically reduced to 73KB by using 2-input and 3-input LUTs with 7 sampling points, instead of using the widely used 4-input LUTs and 17 sampling points settings (Jo & Kim, 2021; Ma et al., 2022; Liu et al., 2023; Li et al., 2024b). Due to the increase in the total number of LUTs, the computational and energy costs result in more than doubled compared to MuLUT-QBD, however, this is within an acceptable criteria for ISP hardware implementation. Note that the most important factor for hardware implementability is the model size, which affects the physical size of the chip.

Although DNN-based methods show better performance, they are much less efficient than LUT-based methods in terms of computation and energy cost. Specifically, PIPNet is more than 100 times more expensive than our method both in memory and computation. These characteristics make efficient hardware implementation very challenging. Notably, our method achieves a very compact LUT size of 73KB while maintaining a reasonable performance. We found that this size is affordable to conventional image signal processor hardware in mobile processors.

**Qualitative Comparisons** Fig. 6 shows the visual comparisons with other methods. MuLUT-BD and MuLUT-QBD show noticeable artifacts due to the absence of quad Bayer adaptive processing. KLAP, being trained on inverse tone-mapped images, shows poor performance on conventional RGB images (train-test distribution mismatch) despite of applying its meta-update feature. PIPNet generally shows good sharpness by employing extensive computations. However, while PIPNet shows poor performance in specific areas such as thin edges, the proposed method achieves better results in terms of noise and aliasing artifacts despite using a much smaller model and fewer computations, through the directional chroma processing. We conjecture it's because the proposed gradient block can effectively process image gradients through directionally stacked LUTs.

Please refer to the appendix A for ablation study, failure case, and analysis on sampling points.

## 5 CONCLUSION

This study presents the first demonstration of LUT-based quad bayer demosaicing. By proposing a residual LUT approach, we reduce the LUT size to make it suitable for hardware implementation. We effectively increased the receptive field by stacking multiple LUTs both in serial and parallel. Furthermore, we apply a directional chroma processing, inspired by traditional demosaicing methods, to mitigate artifacts. While this paper focused only on the G color channel, we believe it can be extended to RB channels with a slightly modified model design.

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

## A APPENDIX

**Slash and Backslash Directional Processing**

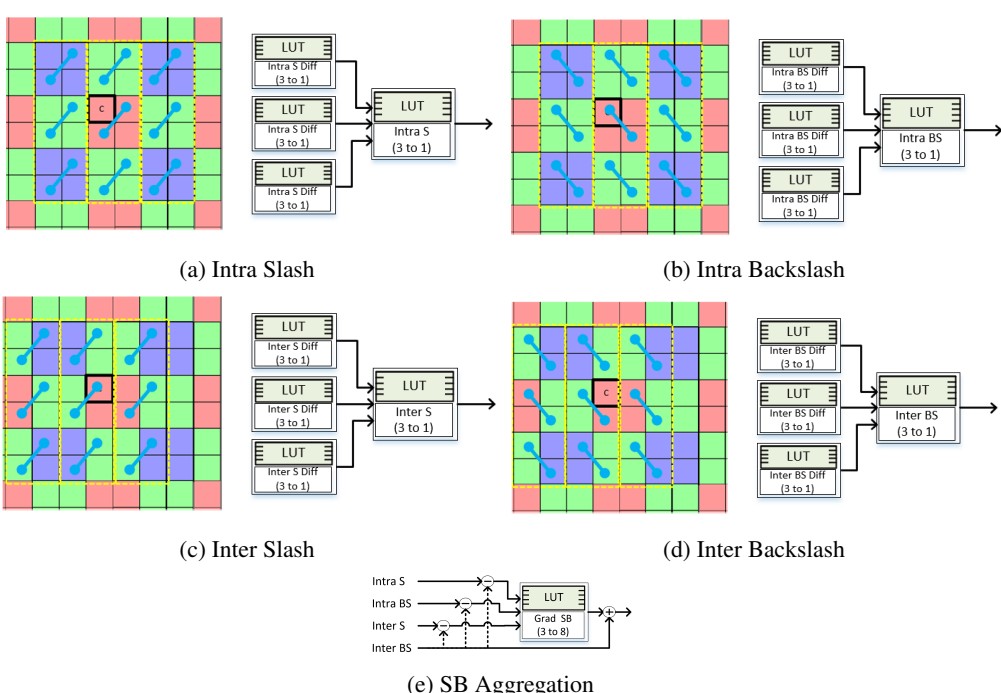

(a) Intra Slash

(b) Intra Backslash

(c) Inter Slash

(d) Inter Backslash

(e) SB Aggregation

Figure 7. The slash and backslash directional processing in the gradient block of the first stage.

**Ablation Study**

Table 2. The results of ablation study. The proposed method achieves superior performance through the combination of all the components, while the stepwise finetuning proves to be essential.

| Method | Size (KB) | GIOPs | Energy (mJ) | Kodak PSNR | Kodak SSIM | McMaster PSNR | McMaster SSIM | Urban100 PSNR | Urban100 SSIM | BSD100 PSNR | BSD100 SSIM | MIT Moire PSNR | MIT Moire SSIM |
|---|---|---|---|---|---|---|---|---|---|---|---|---|---|
| w/o 2nd stage | 39.32 | 2.395 | 2.68 | 33.37 | 0.956 | 33.36 | 0.952 | 29.94 | 0.953 | 31.63 | 0.948 | 30.04 | 0.915 |
| w/o Ch&HF blocks | 25.05 | 2.633 | 2.72 | 31.56 | 0.918 | 33.31 | 0.943 | 28.18 | 0.917 | 29.92 | 0.899 | 29.41 | 0.882 |
| w/o Gradient blocks | 48.23 | 2.321 | 2.86 | 36.30 | 0.966 | 36.42 | 0.966 | 34.62 | 0.971 | 34.69 | 0.963 | 32.68 | 0.932 |
| w/o ft. | 73.28 | 4.789 | 5.37 | 35.41 | 0.953 | 35.98 | 0.962 | 33.96 | 0.964 | 33.88 | 0.95 | 32.37 | 0.929 |
| Ours | 73.28 | 4.789 | 5.37 | 37.18 | 0.972 | 36.87 | 0.968 | 35.58 | 0.976 | 35.69 | 0.971 | 33.42 | 0.939 |

| w/o Ch&HF b. | w/o Gradient b. | w/o 2nd stage | w/o ft. | Ours |
|---|---|---|---|---|

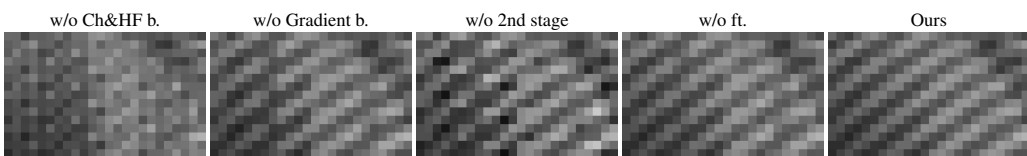

Figure 8. The qualitative results of ablation study. Our method effectively handles aliasing-prone areas caused by the quad Bayer pattern (2px-wide repeated lines).

We conduct an ablation study to verify the effectiveness of each part of our method, and the result is shown in Table 2 and Fig. 8. From the results, we confirm that performance degradation occurs when each proposed component is removed, in order from the first row: the second stage, the chroma and high-frequency blocks, the gradient blocks, and the stepwise finetuning. This verifies that optimal performance is achieved when all the components are combined, proving that each component is essential. Especially, the second stage is necessary to address the most severe visual artifacts,

which are difficult to resolve within the single stage processing. Note that end-to-end simultaneous finetuning decreases the overall performance (PSNR 36.63 on Kodak) compared to our stepwise finetuning due to our multiple path LUT pipeline.

**Sampling Points Variations**

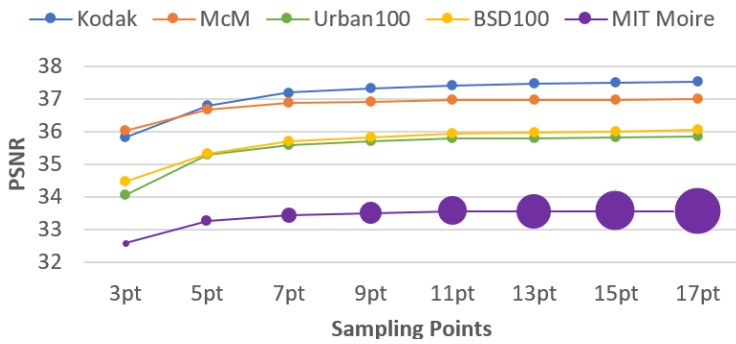

Figure 9. The result of varying sampling points. Our method use 7 points because the saturation becomes severe around it. The area of the purple circle shows a relative comparison of the LUT size according to the sampling points.

We examine the relationship between LUT size and PSNR performance by varying sampling points from 3 to 17. As shown in Fig. 9, the most significant saturation occurs around 7-point. The purple circles are shown to provide a relative comparison of LUT sizes for each sampling point. Considering both LUT size and PSNR performance comprehensively, we confirm that 7-point is suitable configuration for efficient hardware implementation in smartphones or edge devices. Note that the LUT size and the energy cost remain constant regardless of the sampling points.

**Failure Case**

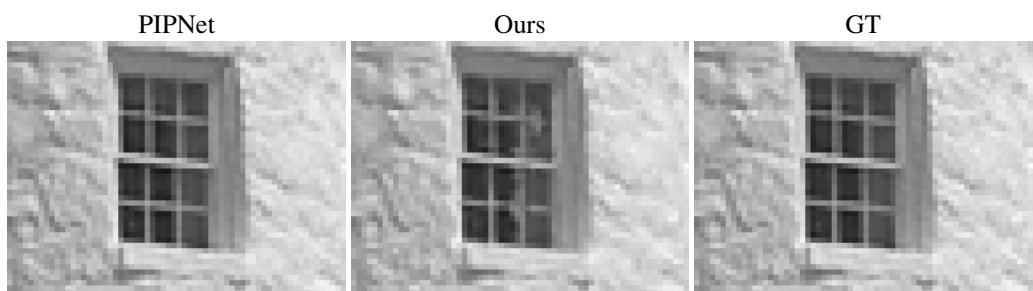

Figure 10. A failure case. We found that artifacts sometimes occur in corner areas.

Fig. 10 shows a failure case of our method. Traditionally, one of the difficult parts in the demosaicing task is corners, and we found that our method occasionally causes artifacts due to the lack of corner-aware processing.

