# OpenReview forum: "Efficient Quad Bayer Demosaicing with Look-Up Tables"
_ICLR.cc/2026/Conference — ICLR 2026 Conference Withdrawn Submission_

### Official Review · Reviewer_Nk16 · 2025-10-30

**Soundness:** 2
**Presentation:** 3
**Contribution:** 2
**Rating:** 2
**Confidence:** 4

**Summary:**

This paper presents a novel, lookup table (LUT)-based approach for Quad Bayer demosaicking, targeting efficient computation and hardware-friendly implementation. The proposed method adopts a two-stage architecture: the first stage performs initial demosaicking, while the second stage focuses on enhancing high-frequency components to restore fine details and edges. Each stage is composed of multiple residual LUTs, which are flexibly stacked in both serial and parallel configurations to expand the effective receptive field without significantly increasing computational cost. Notably, the model achieves competitive image quality even with a very compact LUT size (as small as 73KB), making it highly suitable for deployment in resource-constrained environments such as mobile ISPs. Experimental results demonstrate that the method outperforms existing approaches in terms of efficiency and maintains robust performance across different sensor data.

**Strengths:**

- The paper introduces the LUT-based deep learning framework tailored specifically for Quad Bayer demosaicking. This rethinking of neural network components as compact, reusable lookup tables is conceptually novel and aligns well with hardware optimization goals.
- With a total LUT size of only 73KB, the method achieves strong performance while being exceptionally lightweight. This makes it ideal for real-time applications and embedded systems where memory and power are limited.
- The separation between coarse reconstruction (Stage 1) and detail refinement (Stage 2), particularly focusing on high-frequency enhancement, allows for better control over image quality and edge preservation.
- The paper includes evaluations demonstrating performance gains over baseline methods, supporting claims about efficiency and effectiveness.

**Weaknesses:**

- The paper does not actually perform demosaicking. It only reconstructs the green (G) channel with a 50% sampling rate. This cannot be regarded as a demosaicking task, since true demosaicking requires the estimation of all three RGB channels. Moreover, effective demosaicking methods typically exploit inter-channel correlations, which are not utilized in the current approach.

- The experiments are conducted on clear scenes, while the Quad Bayer pattern was originally designed for low-light conditions, where pixel binning is applied to improve the signal-to-noise ratio (SNR) at the cost of spatial resolution.

- The paper lacks validation on real Quad Bayer images. Given that many smartphones equipped with Quad Bayer  CFAs are capable of capturing RAW images, the proposed method should be evaluated on real-world data to demonstrate its practical effectiveness.

- The paper does not test the inference speed and throughput achieved on actual hardware (e.g., FPGA or mobile NPU).

**Questions:**

-  From an experimental perspective, this paper is far from complete. Please refer to the weaknesses described above.
- Can the current framework adapt to varying CFA patterns or must it be retrained for each specific sensor layout? What is the transferability across devices?
- What are the inference speed and throughput achieved on actual hardware (e.g., FPGA or mobile NPU)? Providing real-world latency metrics would greatly enhance the practical relevance of the claims.

---

### Official Review · Reviewer_e7gE · 2025-11-01

**Soundness:** 2
**Presentation:** 2
**Contribution:** 2
**Rating:** 4
**Confidence:** 2

**Summary:**

This paper introduces an efficient method for Quad Bayer pattern demosaicing using look-up tables (LUTs). Addressing the high computational cost of existing deep learning-based methods, the authors propose a pipeline centered around a new "residual LUT" concept, which significantly reduces the memory and computational complexity. The proposed architecture consists of a two-stage process: a primary demosaicing stage followed by a high-frequency enhancement stage. To handle the large receptive field required for demosaicing, the method stacks multiple small residual LUTs in both serial and parallel configurations. By integrating principles from classical demosaicing, such as directional chroma processing, the model achieves a good balance between performance and efficiency. Experiments show that the method achieves competitive image quality and significantly lower energy cost compared to DNN-based approaches, making it highly suitable for hardware implementation on edge devices.

**Strengths:**

S1. The paper introduces a practical LUT-based approach for Quad Bayer demosaicing. The core "residual LUT" concept is a clever way to reduce a 4D LUT to a 3D LUT, fundamentally slashing memory and computational requirements.

S2. The efficiency of the proposed approach is highly compelling for mobile/edge hardware.

S3. The two-stage design (primary demosaicing + high-frequency enhancement) is reasonable. The use of cascaded LUTs (both serial and parallel) to build a large receptive field from small components is a good design choice.

**Weaknesses:**

W1. Demosaicing is a color reconstruction task. By addressing only the green channel, the paper fails to solve the actual problem. The claim that it "can be extended" to R/B channels is insufficient without experimental results. As presented, the work is a gray-only reconstructor, not a full demosaicing solution.

W2. The method feels more like bespoke, hardware-centric engineering. The architecture (Fig. 4 and 7) is an intricate, hand-crafted cascade of specialized LUTs.

W3. The gradient block design (Fig. 4, Fig. 7) is extremely complex and poorly explained.

W4. The "stepwise finetuning" (Sec 3.5) is presented as an empirical trick "determined through extensive experiments"  with no theoretical or intuitive justification for the specific order, weakening the methodological rigor.

**Questions:**

Refer to Weakness.

---

### Official Review · Reviewer_691u · 2025-11-02

**Soundness:** 2
**Presentation:** 2
**Contribution:** 1
**Rating:** 2
**Confidence:** 4

**Summary:**

The paper introduces a two-stage residual Look-Up Table (LUT)-based framework for quad Bayer demosaicing. It aims for a hardware-efficient, low-latency solution suitable for resource-constrained devices like smartphone ISPs, featuring a compact model size of 73kB.

**Strengths:**

The residual LUT design is genuinely compact and hardware-friendly, addressing a key industry demand, also the need for efficient solutions compared to large Deep Neural Networks (DNNs) is well-articulated.

**Weaknesses:**

Lacks direct, concrete efficiency comparisons (latency/energy) against modern, lightweight CNNs (e.g., MobileNetV3). The method is not demonstrated on the full Red (R) and Blue (B) channels, limiting its practical applicability for full-color images. Lacks deep analysis of ablation results (e.g., why the 2nd stage is necessary; how the design leverages the quad-Bayer pattern). PSNR gains reported do not consistently translate to better visual sharpness than rival methods (e.g., PIPINet).

**Questions:**

The core idea is promising for industrial application, but the submission is fundamentally limited by insufficient experimental validation, limited scope (green-only), and a lack of detailed analysis. The authors must provide stronger evidence of efficiency against true state-of-the-art baselines and demonstrate full-color demosaicing capability.

---

### Note · Authors · 2025-11-20

I have read and agree with the venue's withdrawal policy on behalf of myself and my co-authors.